# i-Sim2Real: Reinforcement Learning of Robotic Policies in Tight Human-Robot Interaction Loops

**Saminda Abeyruwan**\*, **Laura Graesser**\*, **David B. D'Ambrosio**, **Avi Singh**,
**Anish Shankar**, **Alex Bewley**, **Deepali Jain**, **Krzysztof Choromanski**, **Pannag R. Sanketi**
Robotics at Google
{saminda,lauragraesser,ddambro,singhavi,phinfinity,bewley,
jaindeepali,kchoro,psanketi}@google.com

**Abstract:** Sim-to-real transfer is a powerful paradigm for robotic reinforcement learning. The ability to train policies in simulation enables safe exploration and large-scale data collection quickly at low cost. However, prior works in sim-to-real transfer of robotic policies typically do not involve any human-robot interaction because accurately simulating human behavior is an open problem. In this work, our goal is to leverage the power of simulation to train robotic policies that are proficient at interacting with humans upon deployment. But there is a chicken and egg problem — how to gather examples of a human interacting with a physical robot so as to model human behavior in simulation without already having a robot that is able to interact with a human? Our proposed method, Iterative-Sim-to-Real (i-S2R), attempts to address this. i-S2R bootstraps from a simple model of human behavior and alternates between training in simulation and deploying in the real world. In each iteration, both the human behavior model and the policy are refined. For all training we apply a new evolutionary search algorithm called Blackbox Gradient Sensing (BGS). We evaluate our method on a real world robotic table tennis setting, where the objective for the robot is to play *cooperatively* with a human player for as long as possible. Table tennis is a high-speed, dynamic task that requires the two players to react quickly to each other's moves, making for a challenging test bed for research on human-robot interaction. We present results on an industrial robotic arm that is able to cooperatively play table tennis with human players, achieving rallies of 22 successive hits on average and 150 at best. Further, for 80% of players, rally lengths are 70% to 175% longer compared to the sim-to-real plus fine-tuning (S2R+FT) baseline. For videos of our system in action please see https://sites.google.com/view/is2r.

**Keywords:** sim-to-real, human-robot interaction, reinforcement learning

## 1 Introduction

Sim-to-real transfer has emerged as a dominant paradigm for learning-based robotics. Real world training is often slow, cost-prohibitive, and poses safety-related challenges, so training in simulation is an attractive alternative and has been explored for a number of real world tasks, including object manipulation [1, 2, 3, 4], legged robot locomotion [5, 6], and aerial navigation [7, 8]. However, one element that is missing in this prior work is that the policies are not trained to be proficient at interacting with humans upon deployment. The utility of sim-to-real learning can be greatly increased if we extend it to settings where the trained policies need to interact with humans in a close, tight-loop fashion upon deployment. One of the major promises of learning-based robotics is to deploy robots in human-occupied settings, since non-learning robots already work well in deterministic, non-human occupied settings, such as factory floors. However, simulating human behavior is non-trivial (and indeed, one of the primary goals of artificial intelligence research), making it a major bottleneck in sim-to-real research for tasks involving human-robot interaction.

One approach to simulating human behavior is imitation learning. Given a few examples of human behavior, we can use techniques such as behavior cloning [9, 10], or inverse reinforcement learning [11, 12] to distill that behavior into a policy, and then use these policies to generate human

---

\*Indicates equal contribution.

6th Conference on Robot Learning (CoRL 2022), Auckland, New Zealand.

behavior in simulation. However, this approach presents a chicken and egg problem: in order to obtain useful examples of human behavior (in the context of human-robot interaction), we need a robot policy that already knows how to interact with humans in the real world, but we cannot learn such a policy without the ability to simulate human behaviors in the first place. The primary contribution of this paper is a practical solution to this problem.

Our proposed method involves learning a coarse model of human behavior from initial data collected in the real world to bootstrap reinforcement learning of robotic policies in simulation. Deploying this learned policy in the real world now allows us to collect data in which the human subjects meaningfully interact with the robot. We then use this real world experience to improve our human behavior model, and continue training the robot policy in simulation under this updated model. We repeat this iterative process until a desired level of performance is achieved.

We present results on a task involving a robot playing table tennis with non-professional human players (see Figure 1). The goal for the robot is to maximize rally length, i.e. the number of successive hits by the robot and human before the ball goes out of play and policies are evaluated using rally length. Table tennis is a high-speed, dynamic task that requires close, tight-loop interactions between two players (in this case, a human and a robot). Further, maximizing rally length requires the robot to *cooperate* with a human, and vice versa. Thus we believe it to be a good instantiation of our problem setting. We build an initial model of the human player's ball trajectories without a robot present and iteratively refine the robot and player models as they play together, ultimately resulting in a robot policy that can hold rallies of 22 successive hits on average and 150 at best.

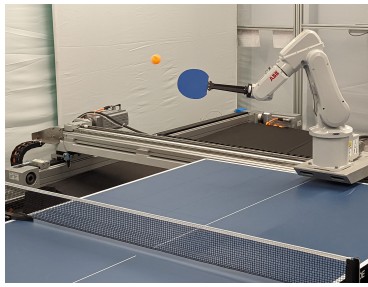

Figure 1: **Robot setup** An ABB IRB 120T 6-DOF robotic arm is mounted to a two-dimensional Festo linear actuator, creating an 8-DOF system.

While we demonstrate our approach on table tennis, we believe that our overall pipeline can be applied to a broad range of tasks, and take into account the various nuances of those tasks. The two characteristics a human behavior model needs to be compatible with our approach are **(a)** it can be updated using human data that is gathered whilst a human or humans are interacting with a robot, and **(b)** the model can be used to sample human behavior in simulation.

In summary, the primary contributions of this paper are: **(a)** a framework for training robotic policies in simulation that would need to interact with human subjects upon deployment, **(b)** a real world instantiation of this framework on a high-speed, dynamic task requiring tight, closed-loop interactions between humans and robots, **(c)** a detailed assessment of how our method, which we call Iterative-Sim-to-Real (i-S2R), compares with a baseline sim-to-real approach in the domain of cooperative robotic table tennis, and **(d)** the first robotic table tennis policy trained to control robot joints using reinforcement learning (RL) that can handle a wide variety of balls and can rally consistently with non-professional humans. i-S2R can apply any RL method, however the only policy-optimization algorithm that so far led to the on-robot-deployable policies is the so-called *Blackbox Gradient Sensing* (BGS) that we introduce here. To see videos of our system in action, please see the supplementary materials and https://sites.google.com/view/is2r.

## 2 Related Work

**Sim-to-Real Learning for Robotics** RL is a powerful paradigm for learning increasingly capable and robust robot controllers [13, 14, 15]. However, learning controllers from scratch on a physical robot is often prohibitively time consuming due to the large number of samples required to learn competent policies and potentially unsafe due to the random exploration inherent in RL methods [16, 17]. Training policies in simulation and transferring them to a physical robot, known as sim-to-real transfer (S2R), is therefore appealing.

Whilst it is both fast and safe to train agents from scratch in simulation, S2R presents its own challenge — persistent differences between simulated and real world environments that are extremely difficult to overcome [17, 18]. No single technique has been found to bridge the gap by itself. Instead a combination of multiple techniques are typically required for successful transfer. These include system identification [13, 19, 20, 21, 22] which may involve iterating with a physical robot in the loop [2, 23], building hybrid simulators with learned models [5, 13, 22], dynamics randomization [1, 2, 5, 6, 13, 14, 15], simulated latency [15, 22], and more complex network architectures [13]. We use

(1) system ID with a physical robot in the loop, (2) dynamics randomization, (3) simulated latency, and (4) more complex networks. Similarly to Lee et al. [13], we use a 1D CNN to represent control policies. Yet a sim-to-real gap persists. Continuing to train in the real world [24, 25, 26, 27] (known as fine-tuning) is an effective way to bridge the remaining gap since the policy can adapt to changes in the environment. We also utilize fine-tuning in this work, but unlike most past work, our learned policy is expected to interact cooperatively with a real human during this fine-tuning phase.

The closest sim-to-real approaches in prior work are Chebotar et al. [2] and Farchy et al. [23] since they update simulation parameters based on multiple iterations of real world data collection interleaved with simulated training. However, both of these prior works focus on using real world interaction data to learn improved *physical* parameters for the simulator, whereas our method focuses on learning better human behavior models. Unlike these prior works, our learned policies are proficient at interacting with humans upon deployment in the real world.

**Reinforcement Learning for Table Tennis**  Robotic table tennis is a challenging, dynamic task [28] that has been a test bed for robotics research since the 1980s [29, 30, 31, 32, 33]. The current exemplar is the Omron robot [34]. Until recently, most methods tackled the problem by identifying a virtual hitting point for the racket [35, 36, 37, 38, 39, 40, 41, 42]. These methods depend on being able to predict the ball state at time $t$ either from a ball dynamics model which may be parameterized [35, 36, 43, 44] or by learning to predict it [33, 38, 39]. This results in a target paddle state or states and various methods are used to generate robot joint trajectories given these targets [33, 35, 36, 43, 44, 45, 46, 47, 48, 49, 50]. More recently, Tebbe et al. [51] learned to predict the paddle target using RL.

An alternative line of research seeks to do away with hitting points and ball prediction models, instead focusing on high frequency control of a robot's joints using either RL [28, 39, 52] or learning from demonstrations [46, 53, 54]. Of these, Büchler et al. [28] is the most similar, training RL policies to control robot joints from scratch at high frequencies given ball and robot states as policy inputs. However Büchler et al. [28] restricts the task to playing with a ball thrower on a single setting, whereas we focus on the harder problem of cooperative play with different humans.

Most prior work simplifies the problem by focusing on play with a ball thrower. Only a few [46, 49, 51, 55] focus on cooperative rallying with a human. Of these, Tebbe et al. [51], is the most similar, evaluating policies on various styles of human-robot cooperative play. However, Tebbe et al. [51] simplify the environment to a single-step bandit and the policy learns to predict the paddle state given the ball state at a pre-determined hit time $t$. In contrast, we learn closed-loop policies that operate at a high frequency (75Hz), removing the need for a learned policy to accurately predict where the ball will be in the future, increasing the robustness of the system, and enabling more dynamic play.

**Human Robot Interaction**  Although not a typical HRI benchmark, cooperative robotic table tennis exhibits many of the features studied in the field: a human and robot working together, complex interactions between the two, inferring actions based on non-explicit cues, and so on. A major challenge in HRI is effectively modeling the complexities of human behavior in simulation [56] in order to learn without requiring an actual human. We employ several common techniques from HRI to learn in simulation such as simplifying the human model [57], specialized models for specific players [58], and refining our model based on real world interactions. Finally we note that like us, Paleja et al. [59] found policy performance varied depending on the skill of the human player.

## 3  Preliminaries

**Problem Setting**  We consider the problem of cooperative human-robot table tennis as a single-agent sequential decision making problem in which the human is a part of the environment. We formalize the problem as a *Markov Decision Process* (MDP) [60] consisting of a of a 4-tuple ($\mathcal{S}$, $\mathcal{A}$, $\mathcal{R}$, $p$), whose elements are the state space $\mathcal{S}$, action space $\mathcal{A}$, reward function $\mathcal{R} : \mathcal{S} \times \mathcal{A} \to \mathbb{R}$, and transition dynamics $p : \mathcal{S} \times \mathcal{A} \to \mathcal{S}$. An episode ($s_0, a_0, r_0, ..., s_n, a_n, r_n$) is a finite sequence of $s \in \mathcal{S}$, $a \in \mathcal{A}$, $r \in \mathcal{R}$ elements, beginning with a start state $s_0$ and ending when the environment terminates. We define a parameterized policy $\pi_\theta : \mathcal{S} \to \mathcal{A}$ with parameters $\theta$. The objective is to maximize $\mathbb{E}\left[\sum_{t=1}^{N} r(s_t, \pi_\theta(s_t))\right]$, the expected cumulative reward obtained in an episode under $\pi_\theta$.

We make two simplifications to our problem. First, we focus on rallies starting with a hit instead of a table tennis serve to make the data more uniform. Second, an episode consists of a single ball throw and return. Policies are therefore rewarded based on their ability to return balls to the opposite side of

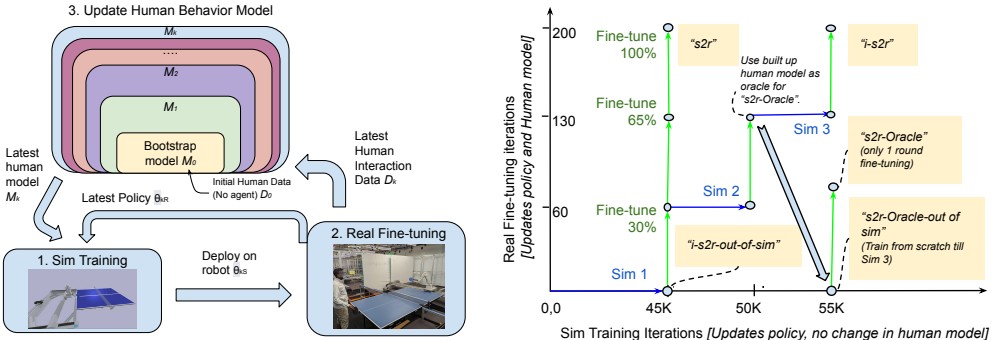

Figure 2: **Iterative-Sim-to-Real. left** We start with a coarse bootstrap model of human behavior (shown in yellow), and use it to train an initial robot policy in simulation. We then fine-tune this policy in the real world against a human player, and the human interaction data collected during this period is used to update the human behavior model used in simulation. We then take the fine-tuned policy back to simulation to further train it against the improved human behavior model, and this process is repeated until robot and human behaviors converge. **right** Specific i-S2R details used in this work. $x$-axis represents the training iterations in sim, $y$-axis represents the fine-tuning iterations in real with human-in-the-loop. Model names are in *italics*.

the table. This reward structure encourages longer rallies, as an agent that can return any ball can also rally indefinitely provided the simulated single shots overlap with the real rally shots.

**BGS & Evolutionary Search (ES)**  i-S2R is compatible with any RL algorithm. In initial experiments we tried a range of methods — PPO [61], QT-OPT [62], SAC [63], and Blackbox Gradient Sensing (BGS) that we introduce here. Only BGS transferred well to a physical robot, hence we continued with this approach and we leave to future work more exhaustive research on other RL algorithms. BGS is an ES-method [64, 65, 66, 67, 68, 69] which have been shown to be an effective strategy for solving MDPs [66, 68]. ES methods aim to optimize the smoothened version $F_\sigma(\theta)$ of the original RL-objective $F(\theta)$, where $\theta$ stands for the policy parameters, given (for the parameter $\sigma > 0$) as:

$$F_\sigma(\theta) = \mathbb{E}_{\delta \sim \mathcal{N}(0, \mathbf{I}_d)}[F(\theta + \sigma\delta)]. \tag{1}$$

Different ES algorithms apply different Monte-Carlo strategies to approximate the gradient of $F_\sigma(\theta)$. In BGS, following [64] we choose Monte Carlo samples $\delta_i$ to form orthogonal-ensembles (to reduce the variance of the estimation) and apply a novel technique for choosing a final collection of samples $\delta_i$ for gradient estimation (the so-called *elite-choice process*). The former technique improved convergence in training and the latter was crucial for the overall effectiveness of training — training in simulation failed without it. See Appendix B for details.

## 4   Method

i-S2R consists of two core components: **(1)** an iterative procedure for progressively updating and learning from a human behavior model — the human ball distribution in this setting — and **(2)** a method for modeling human behavior in simulation given a dataset of human play gathered in the real world (see Figure 2 for an overview). We first describe our iterative training procedure, and then discuss how we model human ball distributions.

**Iterative Training Procedure**  An overview of the method can be seen in Figure 2. First we gather an initial dataset, $D_0$, from player $P$ hitting table tennis balls across the table without a robot doing anything. From $D_0$, we build our first human behavior model $M_0$ that defines a ball distribution (see below). A robot policy is trained in simulation to return balls sampled from $M_0$. Once the policy has converged, we transfer the parameters, $\theta_{0S}$, to a real robotic system. The model is fine-tuned whilst player $P$ plays cooperatively (i.e. trying to maximize rally length) with the robot for a fixed number of parameter updates to produce $\theta_{0R}$. All of the human hits during this fine-tuning phase are added to $D_0$ to form $D_1$, which is used to define $M_1$. The policy weights, $\theta_{0R}$, are then transferred back to simulation and training is continued with the new distribution $M_1$. After training in simulation, the policy weights $\theta_{1S}$ are transferred back to the real world. The fine-tuning process is repeated to produce the next set of policy parameters $\theta_{1R}$, dataset $D_2$, and human model $M_2$. This process can be repeated as many times as needed.

A useful check for assessing convergence was found by looking at the delta in our human behavior model from one iteration to the next. We found the delta between $M_1$ and $M_2$ was substantially smaller than between $M_0$ and $M_1$ indicating that three iterations were enough for this task. For details on the ball distribution parameters for different players see subsection C.3.

**Modeling Human Ball Distributions**  One of our primary goals is to simulate human player behaviors from a set of real world ball trajectories that have been subjected to air drag, gravity, and spin. Due to perception challenges in the real world, we do not explicitly model spin. The input to this procedure is a dataset of ball trajectories, where each trajectory consists of a sequence of ball positions. The output is a uniform ball distribution defined by 16 numbers: the minimum and maximum initial ball position (6), velocity (6), and $x$ and $y$ ball landing locations on robot side (4).

The ball distribution is derived from the dataset in two stages. The first step is to estimate a ball's initial position and velocity for each trajectory. We do this by selecting the free flight part of the trajectory (before the first bounce) and minimize the Euclidean distance between the simulated and real trajectory using the Nelder-Mead method [70]. Please see subsection C.4 for details on the model used to simulate a ball trajectory.

Next we remove outliers using DBSCAN [71] and take the minimum and maximum per dimension to define the ball distribution. We sample an initial position and velocity from this distribution and generate a ball trajectory in simulation subject to the drag force. Other parameters needed for the simulation, such as the coefficient of restitution, friction between the table and ball and the robot paddle and the ball, and so on have been empirically estimated following [72, 73].

## 5    System, Simulation, and MDP Details

Our real world robotic system (see Figure 1) is a combination of an ABB IRB 120T 6-DOF robotic arm mounted to a two-dimensional Festo linear actuator, creating an 8-DOF system, with a table tennis paddle mounted on the end-effector. The 3D ball position is estimated via a stereo pair of Ximea MQ013CG-ON cameras from which we process 2D detections, triangulate to 3D, and filter through a 3D tracker. See Appendix D for more details. We concatenate the ball position with the 8-DOF robot joint angles to form an 11-dimensional observation space. Along with the current observation, we pass the past seven observations (a state space of $8 \times 11$) as the input to the policy. The policy controls the robot by outputting eight individual joint velocities at 75Hz. Following Gao et al. [52] we use a 3-layer 1-D dilated gated convolutional neural network as our policy architecture. Details of the policy architecture can be found in Appendix E.

Our simulation is built on the PyBullet [74] physics engine replicating our real environment. We use PyBullet to model robot and contact dynamics whilst balls are modeled as described in section 4. We add random uniform noise of $2\times$ the diameter of a table tennis ball to the ball observation per timestep to aid transfer to a physical system. We also found it necessary to simulate sensor latency, otherwise sim-to-real transfer completely failed. Robot actions as well as ball and robot observation latencies are modeled as parameterized Gaussians based on measurements from the real system. Policies are rewarded for hitting balls and for returning balls in a cooperative manner. See Appendix G for details.

## 6    Experimental Results

**Experimental Setup**  To evaluate our method, we completed the procedure described in section 4 for five different non-professional table tennis players, thus training five independent i-S2R policies. We compare i-S2R with two baselines. First, the standard sim-to-real (S2R) baseline in which a policy is transferred zero-shot from simulation [1, 3, 5, 6, 7, 8]. Second, a stronger baseline of S2R plus fine-tuning (S2R+FT) in which a policy is transferred in simulation and training is continued in the real world. For fair comparison, S2R+FT is given the same real world training budget as i-S2R. We follow the approach in [24] using the same training algorithm throughout and implement an automatic reset for autonomous training. Finally, each player trained a S2R-Oracle+FT policy which was trained in simulation on the penultimate human behavior model obtained through i-S2R and fine-tuned in the real world for 35% of the i-S2R training budget. This is equivalent to the last round of fine-tuning for i-S2R. (See Figure 2 **right**). S2R-Oracle+FT is intended to isolate the effect of the human behavior modeling on final performance, enabling us to better understand what aspects of the i-S2R process matter. Each policy was evaluated by the model's trainer. Select policies were cross-evaluated by two other players. All policies were tested in random order and the identity of the model was kept hidden from the evaluator (*"blind eval"*). Further details can be found in Appendix H.

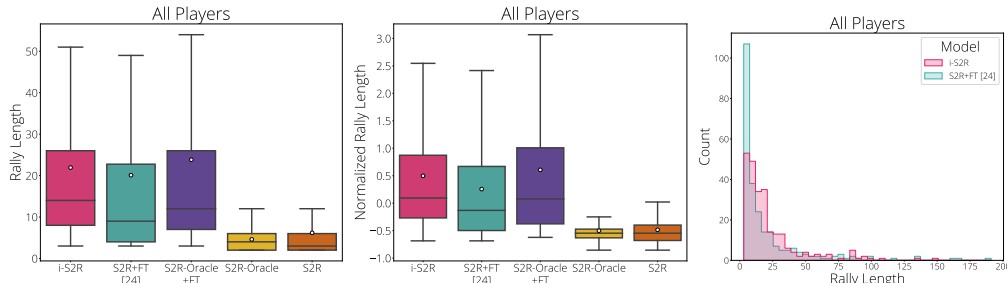

Figure 3: **Aggregated results** *Boxplot details:* white circle: mean, horizontal line: median, box bounds: 25th and 75th percentiles. **left** When aggregated across all players, i-S2R rally length is higher than S2R+FT by about 9%. However, note that simple aggregation puts extra weight on higher skilled players that are able to hold a longer rally. **center** The normalized rally length distribution (see Appendix J for normalization details) shows a bigger improvement between i-S2R and S2R+FT in terms of the mean, median and 25th and 75th percentiles. **right** The histogram of rally lengths for i-S2R and S2R+FT (250 rallies per model) shows that a large fraction of the rallies for S2R+FT are shorter (i.e. less than 5), while i-S2R achieves longer rallies more frequently.

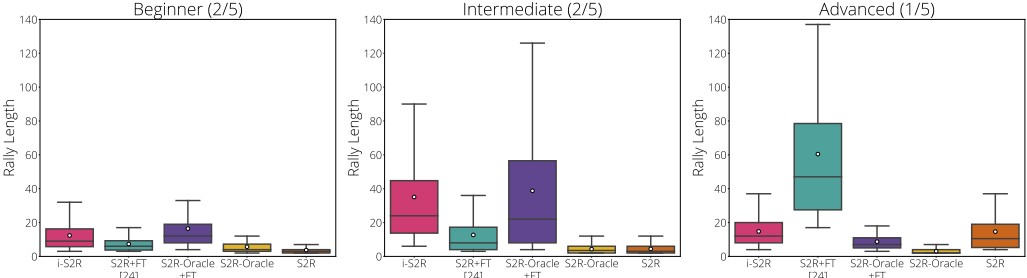

Figure 4: **Results by player skill.** When broken down by player skill, we notice that i-S2R has a substantially longer rally length than S2R+FT and is comparable to S2R-Oracle for beginner and intermediate players. The advanced player is an exception. Note, S2R-Oracle+FT gets just 35% of i-S2R and S2R+FT fine-tuning budget.

Due to the time needed to train and evaluate i-S2R, S2R+FT, and S2R-Oracle+FT (roughly 20 hours per person) we note that 4 of the 5 players are authors on this paper. The non-author player's results appear consistent with our overall findings (see Appendix K for details).

**(1) Does i-S2R improve over S2R+FT in a human-robot interactive setting?** Figure 3 presents rally length distributions aggregated across all players whilst Figure 4 splits the data by skill. Players are grouped into beginner (40% players), intermediate (40% of players) and advanced (20% players). The non-author player was classified as beginner. Please see Appendix I for skill level definitions. When aggregated over all players, we see that i-S2R is able to hold longer rallies (i.e. rallies that are longer than length 5) at a much higher rate than S2R+FT, as shown in Figure 3. When the players are split by skill level, i-S2R significantly outperforms S2R+FT for both beginner and intermediate players (80% of the players). The improvement differs between the two groups, with i-S2R yielding a $\approx 70\%$ and $\approx 175\%$ improvement for beginner and intermediate players respectively.

The policy trained by the advanced player has a different trend. Here, S2R+FT dramatically outperforms i-S2R. We hypothesize that a good S2R model plays a large part in the strong performance of S2R+FT since better transfer from simulation improves the efficiency of subsequent fine-tuning (see Figure 5). One possible explanation for the poor performance of i-S2R is that the policy played fast. During evaluations, we observe the initial robot return is fast with top spin, likely due to a combination of changes in the behavior model from iteration 1 to 2 and 3 and inherent randomness in the training process. In response, the advanced player returns the ball even faster, also with top spin. This appears challenging for the robot to return. During evaluation, most of the errors are made by the robot, where the rally ends with the ball going over the human player's end of the table. This suggests that fine-tuning was not able to adjust in time to the top spin and fast speed of play, causing the robot to hit over the table. One way to mitigate this would be to model spin in simulation, so the policy could learn to respond to spin throughout training, not just during fine-tuning. However, due to the time consuming nature of repeating experiments on the physical system it is difficult to fully explain this result, especially since both the training methodology and involvement of humans introduces a high degree of variance.

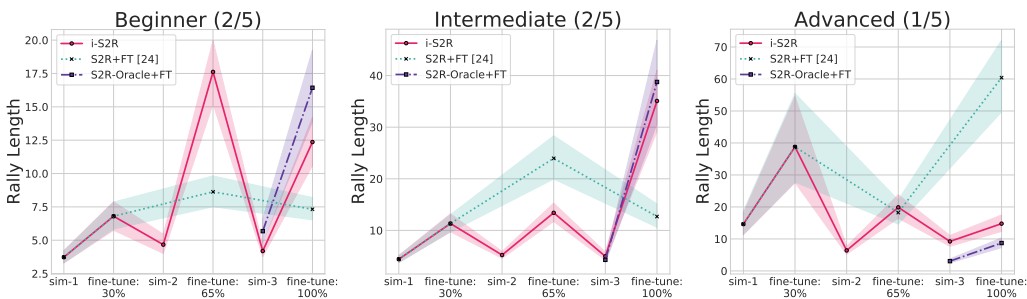

Figure 5: Policy performance at key checkpoints during training. For beginner players i-S2R performance converges after just two iterations (see fine-tune-65%). For intermediate players i-S2R takes three iterations to converge (see fine-tune-100%). "S2R-Oracle-sim-3" here is same as "S2R-Oracle" in Figure 4.

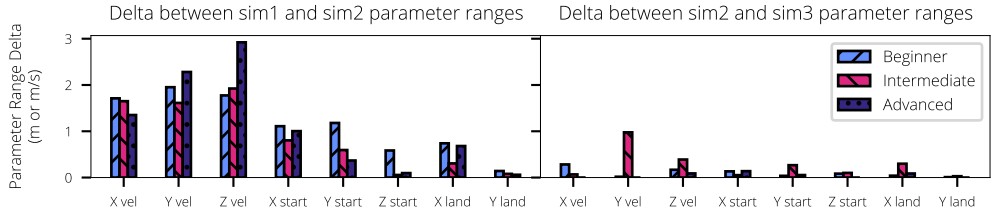

Figure 6: The key distribution parameters change substantially from initial ball distribution (sim1) to that after 1st round of sim training (sim2). This is to be expected given we start from a simple human model (hits across the table). The change in parameters between 1st and 2nd round of sim training is much less (sim2 vs. sim3).

**(2) How many sim-to-real iterations does the human behavior model take to converge?** For beginners we find that it only took two iterations for i-S2R to converge (see Figure 5). In the leftmost chart showing beginner policy data, i-S2R achieves comparable levels of performance at the end of the 2nd (fine-tune-65%) and final (fine-tune-100%) iterations. However, for intermediate skilled players this is not the case. The human behavior model from iteration to iteration (Figure 6) offers a clue. For beginner players, the distribution barely changes after the 2nd round as evidenced by the difference between the left and right charts. Whereas for intermediate players the distribution continues to change substantially from round 2 to 3 (specifically in y and z velocities), which is perhaps why we see the strongest performance of i-S2R after the 2nd iteration for beginners but after the 3rd iteration for intermediate players.

The advanced player's distribution hardly changes between the 2nd and 3rd round and the performance of i-S2R is comparable across both. However this does not explain why we observed the best i-S2R performance at the end of the 1st round for this player. Investigating the effect of playing style on changes in ball distribution every iteration and hence on the sim-to-real gap or training for more iterations for advanced players can shed light on this in future work.

**(3) What is the impact of the human behavior model?** For beginner and intermediate players, S2R-Oracle+FT is in line with i-S2R performance. However S2R-Oracle+FT also achieved this level of performance with just 35% of the real world training time compared to i-S2R and S2R+FT. Therefore much of the benefit of i-S2R likely comes from improving the human behavior model from iteration to iteration. It also suggests that if we had access to the final human behavior model at the beginning of training, the iterative sim-to-real training would not be needed. We could simply fine-tune in the real world and achieve comparable performance with substantially less human training time. S2R-Oracle+FT's strong performance also validates our motivation for this work, in which we hypothesized that the difficulty of defining a good human behavior model a priori for human-robot cooperative rallies was limiting performance.

This result indicates that i-S2R does not benefit from additional training iterations in simulation over and above the improvements to the human behavior model. The evaluations at earlier stages in training (shown in Figure 5) suggest the remaining sim-to-real gap could be responsible. Figure 5 shows that, in all cases, after both the second (sim-2) and third (sim-3) rounds of simulated training, rally length drops noticeably. Reducing the sim-to-real gap might improve i-S2R's performance due to better starting points for the last two rounds of fine-tuning.

**(4) Does i-S2R offer any generalization benefits in this setting?**
We evaluate the generalization capabilities of models trained with
i-S2R, and how they compare against models trained using S2R+FT
by conducting cross-evaluations. A "cross-evaluation" of a policy
is an evaluation conducted by a human who did not play with the
policy during training. Each of the 5 policies was cross-evaluated
by randomly selecting 2 other humans from the human-subject pool
and averaging the results. As shown in Figure 7, i-S2R substantially
outperforms S2R+FT when the models are cross-evaluated by other
players (with similar blind evaluations as earlier) including for the
advanced player where S2R+FT was best in self evaluation (see
Appendix K for details by player). This observation holds whether
we look at absolute or normalized rally length (see Appendix J for
normalization methodology). Performance with other players is
lower for all models, however i-S2R maintains around 70% of per-
formance on average compared to 30% for S2R+FT. We hypothesize
that the broader training distribution obtained by iterating between
simulation and reality leads to policies that can deal with a wider
range of ball throws, leading to better generalization to new players.
Our confidence in this hypothesis is strengthened by the fact that
S2R-Oracle+FT also outperforms S2R+FT in this setting.

Figure 7: Cross-evaluations mean
rally lengths (with 95% CI) ag-
gregated across all players. i-
S2R generalizes better to new
players compared to S2R+FT.

## 7  Limitations

Having a human in the loop poses numerous challenges to robotic reinforcement learning. It slows
down the overall learning process to accommodate human participants, and limits the scale at which
one can experiment. As one example, while we tested our method on five subjects, time limitations
prevented us from training with multiple random seeds for each subject. There is significant variation
in how people interact with robots (or sometimes even the same person over time), which introduces
extra variance into our experiments. In our experiments, the trends we saw for one particular subject
were substantially different from all other subjects, and we could not fully explain why.

It is possible for an expert human player to achieve long rallies by keeping the ball in a very narrow
distribution without really improving the inherent capability of the agent to play beyond those balls.
In our studies, since we used non-professional players, this was not an issue.

Another limitation arising from training a policy with a human in the loop is the possibility that
some performance improvements are attributable to human learning and not policy learning. We did
our best to mitigate this by asking players to evaluate all models "blind" (i.e. the player is unaware
of what model they are evaluating) and at the end of training, after which the majority of human
learning was likely to have occurred. Consequently, we think that differences between models reflect
differences in policy capability and not human capability.

Finally, we represent humans in simulation in a simple way — by capturing all initial position and
velocity ranges during their play — and then we sample each ball in simulation uniformly and
independently. This ignores the probability distribution of balls within those ranges and also results
in a loss of correlation between subsequent balls in a rally. The behavior model also omits spin and
human attributes such as stamina, skill level, intention, and curiosity. These could be addressed by
developing a more sophisticated behavior model that takes these factors into account.

## 8  Conclusion

We present i-S2R to learn RL policies that are able to interact with humans by iteratively training in
simulation and fine-tuning in the real world with humans in the loop. The approach starts with a coarse
model of human behavior and refines it over a series of fine-tuning iterations. The effectiveness
of this method is demonstrated in the context of a table tennis rallying task. Extensive "blind"
experiments shed light on various aspects of the method and compare it against a baseline where we
train and fine-tune in real only once (S2R). We show that i-S2R outperforms S2R in aggregate, and
the difference in performance is particularly significant for beginner and intermediate players (4/5).
Moreover, i-S2R generalizes much better than S2R to other players.

**Acknowledgments**

We thank Pete Florence, Kamyar Ghasemipour, Andrew Silva, Ellie Sanoubari, and Vincent Van-houcke for their helpful and insightful feedback on earlier versions of this manuscript. We are grateful to Michael Ahn, Sherry Moore, Ken Oslund, and Grace Vesom for all their work on the robot control stack, for Omar Cortes' help in training models and for Justin Boyd and Khem Holden's help in calibrating our vision system. We thank Jon Abelian, Gus Kouretas, Thinh Nguyen, and Krista Reymann for all that they do to help maintain our robotic system. We would also like to thank Navdeep Jaitly, Peng Xu, Nevena Lazic, and Reza Mahjourian for their work on early versions of this system. Finally, we would like to thank Jon Abelian, Justin Boyd, Omar Cortes, Khem Holden, Gus Kouretas and Thinh Nguyen for their help evaluating models.

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
