# OpenReview forum: "i-Sim2Real: Reinforcement Learning of Robotic Policies in Tight Human-Robot Interaction Loops"
_robot-learning.org/CoRL/2022/Conference — CoRL 2022 Oral_

### Official Review · Reviewer_LWg8 · 2022-07-01

**Originality:** Good
**Technical Quality:** Good
**Clarity Of Presentation:** Fair
**Impact:** 3

**Recommendation:**

Weak Accept: I recommend accepting the paper, but will not argue for my recommendation if the majority of other reviewers have a different opinion.

**Summary:**

The paper proposed an iterative sim-to-real method for learning to play robotic table tennis. This method trains a policy in simulations, obtains data of rallies between a human and a robot using this policy, and updates the human ball trajectory model using obtained data.
Experimental results indicate that the proposed method outperforms a fine-tuning method and demonstrates the effects of the parameters of the method and levels of a player on the performance of the method.

**Issues:**

- All issues have been resolved.


**Quality Of The Limitations Section:**

Limitations are addressed clearly

**Reviewer Expertise:**

4: The reviewer is confident but not absolutely certain that the evaluation is correct

**Robotics Focus:**

Sufficient demonstration on hardware

**Strengths And Weaknesses:**

**Strengths**

- Introduction describes the background and motivation. This reviewer agrees that it is important for a robot to learn to interact with a human using simulations while updating the simulation using data from trials in the real world.
- Related work section introduces related topics and compares the proposed method with previous methods.
- The sections of the method, system, experiments, and limitations are written clearly and easy to follow.

**Weaknesses**

- All issues have been resolved.

**Summary Of Recommendation:**

The reviewer recommends Weak Accept because the authors resolved all issues that the reviewer concerned.

---

> ### Author Response · Authors · 2022-08-24
> **Authors response to Reviewer LWg8**
>
> **Comment:**
>
> Thank you for your helpful feedback, we appreciate your comments and the time you took to review our work.
>
> **On the evaluation metric**
>
> The goal for the robot is to maximize rally length, i.e. the number of successive hits by the robot and human before the ball goes out of play. Policies are evaluated using this rally length metric. Maximizing rally length requires the robot to cooperate with a human (and vice versa) for a long time (many hits) in a fast, dynamic loop. Thus we believe it to be a good instantiation of our problem setting which is to train robotic policies in tight human-robot interaction loops. We have clarified this in the revised version of our submission (attached, changes highlighted in blue).
>
> **On baseline selection**
>
> As we describe in the related work section, the most common approach to sim-to-real in robotics is zero-shot transfer from simulation [1, 3, 5, 6, 7, 8] and we think this is the standard method. This baseline is included in our results as “i-S2R-out-of-sim” and we acknowledge that this labeling is perhaps confusing. To address this, we have renamed “i-S2R-out-of-sim” to “S2R”.
>
> However, we wished to compare i-S2R with a stronger baseline than the standard method, hence why we mainly compare with sim-to-real plus fine-tuning. This is labeled as “S2R” in our results and we have re-named it to “S2R+FT” to emphasize that fine-tuning is involved. Our implementation of S2R+FT adopts the approach taken [24].
>
> Comparing i-S2R and S2R+FT is a more like-for-like comparison than with S2R because both i-S2R and S2R+FT have the same real world fine-tuning budget. In comparison, S2R does not have any real world training.
>
> We have updated all the charts and text in the revised version of our submission (attached) to reflect these changes.
>
> [1] X. B. Peng, M. Andrychowicz, W. Zaremba, and P. Abbeel.  Sim-to-real transfer of robotic control with dynamics randomization, 2018
>
> [3] M. Andrychowicz, B. Baker, M. Chociej, R. Jozefowicz, B. McGrew, J. Pachocki, A. Petron, M. Plappert, G. Powell, A. Ray, J. Schneider, S. Sidor, J. Tobin, P. Welinder, L. Weng, and W. Zaremba. Learning dexterous in-hand manipulation, 2020
>
> [5] J. Lee, A. Dosovitskiy, D. Bellicoso, V. Tsounis, V. Koltun, and M. Hutter. Learning agile and dynamic motor skills for legged robots, 2019
>
> [6] X. B. Peng, E. Coumans, T. Zhang, T. E. Lee, J. Tan, and S. Levine.  Learning agile robotic locomotion skills by imitating animals, 2020
>
> [7] F. Sadeghi and S. Levine.  CAD2RL: real single-image flight without a single real image, 2017
>
> [8] A. Loquercio, E. Kaufmann, R. Ranftl, M. Muller, V. Koltun, and D. Scaramuzza.  Learning high-speed flight in the wild, 2021.
>
> [24] L. M. Smith, J. C. Kew, X. B. Peng, S. Ha, J. Tan, and S. Levine. Legged robots that keep on learning, 2021
>
> **On the generality of i-S2R**
>
> We would like to point out that our primary contribution in the paper is the i-S2R algorithm: an online, iterative approach that alternates between simulated training, real world training and data-collection, and behavior modeling. We chose a specific approach to model human behaviors for our task, and we do not claim that this particular approach of human behavior modeling will work for any arbitrary task. However, we believe that our overall pipeline of bootstrapping from a small amount of human data, and then iteratively refining both our human behavior models and learned robot policies in an online fashion can be applied to a broad range of tasks, and take into account the various nuances of those tasks. The two characteristics a human behavior model needs to work with i-S2R are (a) it can be updated using human data that is gathered whilst a human or humans are interacting with a robot, and (b) the model can be used to sample human behavior in simulation. We therefore think a large class of human behavior models are compatible with the i-S2R algorithm. Perhaps the most general way to build human models would be to use a powerful generative model (such as an autoregressive, decoder-only transformer), and this direction would be interesting to explore in future work. We have updated the introduction to reflect this discussion.
>
>
> **Zip File:**
>
> /attachment/c6fb25d2909f14561b189f973ebef1f70170ff5b.zip

---

### Official Review · Reviewer_ksqZ · 2022-07-26

**Originality:** Good
**Technical Quality:** Good
**Clarity Of Presentation:** Very Good
**Impact:** 3

**Recommendation:**

Weak Accept: I recommend accepting the paper, but will not argue for my recommendation if the majority of other reviewers have a different opinion.

**Summary:**

This work proposes an iterative training scheme where a policy is trained in simulation and then deployed on a real-robot to fine-tune and collect data from a human player. This iSim2Real system was implemented on a ABB robot arm and demonstrated to collaboratively play table tennis (a rallying task) with beginner and intermediate human players.

**Issues:**

Please see the weaknesses outlined above, and I hope the authors can provide additional discussion around the potential of iS2R to more complex human models and other settings beyond table tennis.

**Quality Of The Limitations Section:**

Additional details required

**Reviewer Expertise:**

4: The reviewer is confident but not absolutely certain that the evaluation is correct

**Robotics Focus:**

Sufficient demonstration on hardware

**Strengths And Weaknesses:**

Positives:
- A big plus of the paper is that the approach was demonstrated and tested on a real-world 8-DOF system on the relatively complex task of table-tennis.
- The iSim2Real framework is simple, but it is not immediately obvious that it would work with real human players.
- Ablation studies are presented showing that the system attains a good policy relatively quickly, along with the impact of improving the human model and play performance across human players.
- The paper is generally well-written and I enjoyed reading it.

Negatives:
- The human model in this setup is relatively simple; it is the ball distribution and the demonstration is limited to this setting. There are no “human aspects” that are directly modeled (e.g., fatigue, expertise, trust) though potentially, this can be captured by the iterative setup at the cost of sim iterations. It would be helpful if the authors discuss this limitation and how the method can be extended towards more complex human models.
- A related issue is that humans change over time, e.g., they can get better over time or can adapt to the robot’s policy. This is not discussed with sufficient depth in the paper but is important in collaborative settings. In the experiments, did the human and robot find a good *joint* policy? Does the i-S2R system facilitate this? Is the system able to adapt to a changing human? Although I understand experiments into these issues may not be possible given the difficulties associated with human-subject experiments, some discussion is required.
- The experimental results are for 5 human subjects (4 of which are the papers authors) and on a single task; it is difficult to draw very strong conclusions from the limited participant pool.
- Something that was unclear in the paper is the key differences between advanced v.s. beginner/intermediate players. Perhaps the authors can analyze the differences between the learnt model distributions? This could also lead to further insights as to why i-S2R did not work well with advanced players.

Minor:
- I did not understand why there are 5 human subjects since each model was evaluated by “(a) the model’s trainer and (b) two other players”.
- it is a little odd that the authors refer to the model as having “converged” since the human may continue to change over time.

**Summary Of Recommendation:**

Overall, the strengths of this paper outweigh the negatives. I especially appreciate that the authors conducted experiments with a real table-tennis robot. The major downsides for me were that the human model (for this setup) is simplified to an extent that the “human” elements appear to be missing, and that the participant pool was limited in size. The latter is somewhat of a missed opportunity; with a larger human-subject experiment, I would have recommended a strong accept.

---

> ### Author Response · Authors · 2022-08-24
> **Authors response to Reviewer ksqZ (1/2)**
>
> **Comment:**
>
> Thank you for your helpful feedback, we appreciate your comments and the time you took to review our work. We’re also glad that you enjoyed reading the paper and appreciate our demonstration of i-S2R on a table tennis task.
>
> **On the simplification of the human model**
>
> We completely agree this is a limitation in the current version of the model. Modeling more human aspects such as fatigue (qualitatively our human players report getting tired as the policies improved), expertise, or trust, as you mention is an exciting direction to pursue. Perhaps the most general way to build human models would be to use a powerful generative model (such as an autoregressive, decoder-only transformer), and this direction would be interesting to explore in future work. Further, were we to incorporate more complex human aspects into the behavior model we would likely need to use a control policy so that it could learn to adapt to the nuances of the behavior model.
>
> However, we do believe that our overall i-S2R pipeline of bootstrapping from a small amount of human data, and then iteratively refining both our human behavior models and learned robot policies in an online fashion can be applied to a broad range of tasks, and take into account the various nuances of those tasks. The two characteristics a human behavior model needs to work with i-S2R are (a) it can be updated using human data that is gathered whilst a human / humans are interacting with a robot, and (b) the model can be used to sample / generate human behavior in simulation. We therefore think that a large class of human behavior models are compatible with the i-S2R algorithm.
>
> We have updated the introduction and limitations section in the revised version of our submission (attached, with changes in blue) to reflect this discussion.
>
> **On the human-subject experiment size**
>
> While our main method only requires 6 hours of in-person fine-tuning, our evaluation involves 3 different fine-tuning runs (our method and two baselines), and we evaluate 10 different checkpoints for each person. In aggregate, adding each person requires an additional 20 hours of human and robot time. This is before we take into account rest periods for humans and any robot downtime that we might encounter. Our experiments already exceeded over 100 hours of human and robot time, and it is difficult and expensive to increase this number in a significant way. Further, since we have only a single robot setup, we have to run our experiments one person at a time, and we are only able to obtain results for a maximum of two people per week (assuming no downtime at all). We do agree that a larger sample size would have been preferable, however given the above constraints we did our best to compensate for the relatively small sample size by making our evaluations as comprehensive and unbiased as possible.
>
> **On the fact that humans change**
>
> We completely agree that humans change over time and that this is important in collaborative robotics. We believe that i-S2R is able to adapt to a changing human — this is precisely what the multiple rounds of real world fine-tuning in i-S2R facilitate. Our focus in this paper was on how to address the fact that a human’s policy changes as the robot gets better. However we think that i-S2R could equally be applied in settings in which a human’s policy changes due to other factors (such as learning or forgetting). Each time the human policy changes an additional iteration could be performed.
>
> We do note the possibility of humans changing during training, specifically of them getting better, in our limitations section (lines 321 - 326 in the original pdf). Since our focus was on evaluating the robots we made efforts to exclude the possibility of humans changing from our evaluations.
>
> Finally, we think that the human and robot do find a good joint policy. Evidence for this is the cross evaluations (Figure 7). Here we see that when a policy is evaluated by a player that wasn’t involved in the training of the policy, performance drops, suggesting some amount of co-adaptation between the robot and player. Qualitatively this was also the experience of our test subjects. They describe learning to adapt to the policy’s style during fine-tuning.
>
> **On model “convergence”**
>
> We agree that humans change, so it is perhaps impossible to say definitively that training has converged. However within the time frame of the experiment, we did observe much less change in the human behavior model with each subsequent iteration (please see Appendix, Table 1). Within the narrower scope of the experiment, we think it is reasonable to say that the human behavior model has converged.
>
> **Zip File:**
>
> /attachment/d834a9f75cdc909cdd9bcdf39ac92a3aff36d6f1.zip

---

> > ### Author Response · Authors · 2022-08-24
> > **Authors response to Reviewer ksqZ (2/2)**
> >
> > **Comment:**
> >
> > **Clarifying “...why there are 5 human subjects since each model was evaluated by “(a) the model’s trainer and (b) two other players”.”**
> >
> > We repeated the i-S2R training process, along with training two baselines, 5 times, each with a different human opponent, resulting in 5 distinct policies per method. Thus there are 5 human subjects and each subject is associated with a distinct policy.
> >
> > A “cross-evaluation” of a policy is an evaluation conducted by a human who was not involved in the training of a policy, i.e. they did not play with the policy during training. We “cross-evaluated” each of the 5 policies by randomly selecting 2 other humans from the human-subject pool. Having all 5 participants evaluate all other participant's models would have required substantially more time from the humans (which was already limited, as mentioned earlier), so we felt this was a fair compromise.
> >
> > We have clarified this in the revised version of our submission.
> >
> >
> > **Zip File:**
> >
> > /attachment/50c7aef104dac330accd15645037848f47e6efc6.zip

---

> > > ### Comment · Reviewer_ksqZ · 2022-08-26
> > > **Thank you for the clear response**
> > >
> > > Thank you for the clear response and for the edits to the paper. The limitations are better discussed now and I can clearly understand the experimental protocol. I acknowledge that more experiments are challenging in this domain and appreciate the authors have done their best given their setup and resources. I do hope that they will consider larger samples in the future to increase the impact of this work.

---

> > > > ### Author Response · Authors · 2022-08-26
> > > > **Thank you for your comments**
> > > >
> > > > We are happy to hear our response and edits improved the limitations and clarified the experimental protocol. Thank you for your additional comments and for all of your feedback. We appreciate it very much!

---

### Official Review · Reviewer_PRVu · 2022-07-28

**Originality:** Very Good
**Technical Quality:** Very Good
**Clarity Of Presentation:** Excellent
**Impact:** 3

**Recommendation:**

Strong Accept: I recommend accepting the paper and will argue for my recommendation even if other reviewers hold a different opinion.

**Summary:**

Enabling robots to imitate human behavior in robot-human interactive domains can be challenging, since a robot policy must already know how to interact with humans in order to obtain useful examples of human behavior to learn from. The work presents a novel method, iIterative-Sim-to-Real (i-S2R), that provides a structured way for a bootstrapped human model prior to be trained in simulation and then further refined in the real world in an iterative loop. The method is evaluated on the table-tennis task across 5 human subjects with varying skill levels, and is shown to outperform a non-iterative variant in most cases, resulting in respectable rally lengths on average (22) and significantly longer rallies (up to 150) as well.

**Issues:**

Questions:
- How dependent is the method of accurate modeling? E.g.: can i-S2R still succeed with less-accurately tuned ball parameters (L189-190)?
- What was the motivation behind choosing 3 fine-tuning stages, vs. additional stages? It does seem necessarily straightforward that the learned human model has “converged” by iteration 3 (which raises another question – what exactly defines “convergence” in this setup?).

Suggestions:
- An ablation study testing a completely “random” human model (fully randomized start pose, vel, landing x, y, etc.) would help validate that humans are actually necessary in the data bootstrapping process, and help contextualize the claim in L277-278.

Technical Concerns:
- The variable scaling in the y-axis between subfigures in Figure 4 seems a bit misleading.
- From what I understand, the initial data from the human used in the first simulation iteration consists solely of single hits without rallying (L164-165). If this is the case, then Fig 6 seems a bit misleading – there is an obvious domain shift between sim1 and sim2 since rallying only starts to occur in sim2+, which may explain the larger delta values seen in that figure.


**Quality Of The Limitations Section:**

Limitations are addressed clearly

**Reviewer Expertise:**

3: The reviewer is fairly confident that the evaluation is correct

**Robotics Focus:**

Sufficient demonstration on hardware

**Strengths And Weaknesses:**

Strengths:
- The model performance on real robot hardware is impressive, achieving respectable rally lengths on average (22) and significantly longer rallies (up to 150) as well.
- The presentation is clear, and details are easy to follow.
- The subject evaluation method (blind comparisons between subjects and trained models) is well-grounded, and removed the potential for biased performance between subjects.
- Overall experimentation is very thorough, with clear motivations for each setup.

Weaknesses:
- The method seems to require strong human prior modeling – i.e.: a distribution that can be both expressed analytically and simulated accurately in simulation. This may be a strong assumption, and could limit the method being generally applied to more complex settings where such modeling might be intractable (e.g.: the cluttered navigation problem described in L61-65 – see related comment below).
- It does not seem clear that the table tennis setup actually represents more ecological tasks, such as the busy hallway navigation task claimed in L61-65. The domain shift seems rather large and nontrivial, especially since the human model would have to account for N other humans (instead of just one), and will probably be in constant “generalization” mode since it is unlikely to have the exact same configuration of humans repeated.
- The advanced player’s data is obviously surprising. The relative outperformance of S2R over i-S2R performance isn’t necessarily concerning (it could perhaps be explained by the consistency of the professional player), but the i-S2R score in comparison with the other plays seems very counterintuitive – suggesting that the method somehow performs better on “worse” players (intermediate, ~40 hit mean rally length) compared to “better” players (advanced, ~20 hit mean rally length).


**Summary Of Recommendation:**

The paper proposes a novel iterative method that exhibits impressive empirical results on real robot hardware. The experimentation method is extensive and thorough, and showcases the strength of the method across varying human subjects. Further discussion on the advanced player’s surprising data, as well as more explicit justification of the applicability of the proposed method to more complex robot-human interactive setups, would further strengthen the core arguments presented in the paper.

---

> ### Author Response · Authors · 2022-08-24
> **Authors response to Reviewer PRVu (1/2)**
>
> **Comment:**
>
> Thank you for your thoughtful and detailed feedback, as well as for your questions and suggestions. We appreciate the time and effort you took to review our work. We’re also glad that you appreciated our results on a physical robot, blind evaluations, and experimentation methodology.
>
> **On the advanced player’s data**
>
> We looked into simulated training curves and did not observe any systematic differences that might explain the surprising performance of i-S2R for the advanced player.
>
> One possible explanation is that the i-S2R policy for the advanced player played fast. During evaluations, we observe the initial robot return is fast with top spin, likely due to a combination of changes in the behavior model from iteration 1 to 2 and 3 and inherent randomness in the training process. In response, the advanced player returns the ball even faster, also with top spin. This appears challenging for the robot to return. During evaluation most of the errors are made by the robot, where the rally ends with the ball going over the human player’s end of the table. This suggests that fine-tuning was not able to adjust in time to the top spin and fast speed of play, causing the robot to hit over the table. One way to mitigate this would be to model spin in simulation, so the policy could learn to respond to spin throughout training, not just during fine-tuning.
>
> We have updated Section 6 (1) and the limitations section of the paper to reflect this discussion. Please see the revised version attached with changes in blue.
>
> **On whether the table tennis setup actually represents more ecological tasks**
>
> We agree that table tennis is different in some important aspects to the hallway navigation task. For example, there might be N humans and M robots instead of 1 human and 1 robot, as you mention. Consequently applying i-S2R to this task may involve using a more complex behavior model. For example, data from multiple humans could be used to build N distinct human behavior models, each of which is sampled from in the environment, and updated with each iteration of i-S2R.
>
> Nevertheless, we do believe that our overall i-S2R pipeline of bootstrapping from a small amount of human data, and then iteratively refining both our human behavior models and learned robot policies in an online fashion can be applied to a broad range of tasks, and take into account the various nuances of those tasks. The two characteristics a human behavior model needs to work with i-S2R are (a) it can be updated using human data that is gathered whilst a human / humans are interacting with a robot, and (b) the model can be used to sample / generate human behavior in simulation. We therefore think that a large class of human behavior models are compatible with the i-S2R algorithm.
>
> If we were to use more complex human behavior models in i-S2R, we would likely need to use a control policy with memory so that it could learn to adapt to the nuances of the behavior model. Such models have been shown to implicitly perform system ID [3], so it is likely they will be able to identify and adapt their behavior to multiple humans represented with a more complex behavior model.
>
> We have updated the introduction in the revised version of our submission to reflect this discussion.
>
> [3] M. Andrychowicz, B. Baker, M. Chociej, R. Jozefowicz, B. McGrew, J. Pachocki, A. Petron, M. Plappert, G. Powell, A. Ray, J. Schneider, S. Sidor, J. Tobin, P. Welinder, L. Weng, and W. Zaremba. Learning dexterous in-hand manipulation, 2020
>
> **Response to Questions**
>
> *What was the motivation between choosing 3 fine-tuning stages vs. additional stages?*
>
> The number of fine-tuning stages was set empirically, by measuring the delta in human behavior model parameter values after each iteration (see Appendix, Table 1). Our initial experiments demonstrated that after the 3rd iteration changes in parameter values were negligible. Therefore, we empirically set the limit to 3 iterations. Having further iterations does not provide additional human behavior information to the model in this setting. In other tasks the optimal number of iterations may vary. We recommend monitoring the delta in the human behavior model from iteration to iteration and stopping when the difference becomes small or when the desired level of performance has been reached. We have clarified this in Section 4 of the revised version of our submission (attached, with changes in blue).
>
>
>
> **Zip File:**
>
> /attachment/2069349fa575402b98dfa2ad1cc160021019d5fc.zip

---

> > ### Author Response · Authors · 2022-08-24
> > **Authors response to Reviewer PRVu (2/2)**
> >
> > **Comment:**
> >
> > *How dependent is the method on accurate modeling?*
> >
> > We endeavored to accurately model the ball in simulation, for example we have measured the physical parameters of the ball, paddle, and table using standard methods, and the simulator is fixed to those values. However, we know that our model is not perfect. For example, we have not modeled spin or physical wear and tear of the balls. These unknown parameters do introduce a sim-to-real gap and we observe a drop-off in performance between simulation and the real world (see Figure 5). Our hypothesis is that fine-tuning is able to adapt reasonably well to these inaccuracies, although we think that were the sim-to-real gap to be reduced, performance would improve.
> >
> > While it is true that i-S2R assumes that the human behavior model can be represented and sampled from in simulation, we note that our model is quite simple. We reduce human ball trajectories into a set of position and velocity bounds which we sample from uniformly. Further, the distribution does not need to be expressed analytically, all that is needed is the ability to sample from it in simulation. A generative model (such as an autoregressive, decoder-only transformer) is also compatible with i-S2R, and this direction would be interesting to explore in future work.
> >
> > For these reasons we believe that the human behavior model requirement is not especially strict and that i-S2R is robust to some inaccuracies in the behavior model. Assessing the sensitivity of the method to different human behavior models of varying degrees of fidelity is an interesting direction for future work.
> >
> > **Response to Suggestions**
> >
> > *Ablation study testing a completely “random” human model.*
> >
> > While we haven’t experimented with a random human behavior model, we ran some initial experiments using a hand-designed human behavior model that we expected to cover a wide variety of human behaviors. When training this model for the same number of iterations as S2R-Oracle, performance was 38% lower for the hand-designed model (Appendix K.1, Table 9 and Figure 17). Since this hand-designed model was significantly worse than the data-driven models, we do not expect a random model to do any better than this.
> >
> > **Response to Technical concerns**
> >
> > Figure 4 variable scaling: We have fixed this in the revised version of our submission.
> >
> > Figure 6: Yes, your understanding is correct. The domain shift is a core aspect of i-S2R, since if we could model the desired human behavior initially then there would be no need for the iterative approach. We have clarified the caption in the revised version of our submission.
> >
> >
> > **Zip File:**
> >
> > /attachment/750bf582e3403204d88309d70d1d9aaf23a50ef8.zip

---

> > > ### Comment · Reviewer_PRVu · 2022-08-26
> > > **Thank for the detailed responses**
> > >
> > > Thank you for the detailed responses -- they provided further insight and helped clarify some questions and confusion I originally had.
> > >
> > > The additional information about the advanced player makes sense; I can imagine spin playing an increasingly crucial role in overall ball trajectory as player skill level increases.
> > >
> > > This actually ties into the discussion on the need for accurate modeling -- it seems that the method's robustness to modeling inaccuracies is not only dependent on the type of inaccuracy (e.g.: ball location vs. spin) but also the severity of the inaccuracy (e.g.: spin becomes more significant as the human's skill increases). I do agree that while intriguing, this is out of the paper's scope and can be left for future work.
> > >
> > > I will update my recommendation to a Strong Accept.

---

> > > > ### Author Response · Authors · 2022-08-26
> > > > **Thank you for your additional comments**
> > > >
> > > > We are happy to hear our response was helpful and clarified your questions. Thank you for your additional comments and for updating your recommendation to a Strong Accept. We appreciate it very much!

---

### Official Review · Reviewer_WKtY · 2022-08-01

**Originality:** Good
**Technical Quality:** Very Good
**Clarity Of Presentation:** Excellent
**Impact:** 4

**Recommendation:**

Strong Accept: I recommend accepting the paper and will argue for my recommendation even if other reviewers hold a different opinion.

**Summary:**

This paper proposes a general methodology for sim-to-real transfer learning for reinforcement learning policies in human-robot interaction scenarios. The method, Iterative-Sim-to-Real (i-S2R), attempts to simultaneously learn a model of both the target policy and the corresponding human behavior in an iterative fashion. An initial estimate of the human behavior is made and used to train the target policy, after which the policy is deployed and fine-tuned in the real world against an actual human. The resulting data is then brought back to simulation to update the human behavior model and the target policy with further training. This process iteratively repeats as needed. This method proposes a solution to the problem of leveraging simulation when humans are involved – how do we model the associated human behavior in simulation? The resulting method is applied to a game of table tennis, and shows improvement over a standard sim-to-real approach.

**Issues:**

* Why is there such a significant difference between s2r-oracle and s2r-oracle-out-of-sim? This seems like it would imply that the human behavior model doesn't actually make that much difference, more that it is the fine-tuning process that makes a significant difference. Are there any results for the intermediate iterations? Results for 45k sim and fine-tune 30% might shed some more light on this, i.e., is it the behavior model or sim that impacts things.
* As noted above in Weaknesses, why is the fine-tuned model from the real-world brought back to simulation? What exactly is being fine-tuned here? One would assume that this is correcting for distribution shifts in the dynamics, but then it wouldn't make sense to bring these changes back to simulation. More information or intuition here would be beneficial.
* Are there statistical significance results for the comparisons between the different S2R variants?
* What distinguishes a beginner, intermediate, and advanced player? This information would be useful to know, either in the paper or appendix.

**Quality Of The Limitations Section:**

Limitations are addressed clearly

**Reviewer Expertise:**

4: The reviewer is confident but not absolutely certain that the evaluation is correct

**Robotics Focus:**

Sufficient demonstration on hardware

**Strengths And Weaknesses:**

Strengths:
* The paper is well-written and well-motivated; the sim-to-real problem being addressed is very relevant to human-robot interaction and so contributions in this area are of great interest to the HRI community.
* The iterative training methodology which interleaves real-world policy rollouts with simulated data to improve the fidelity of the human behavior model is fairly novel and interesting. While the concept of interleaving real-world policy rollouts itself is not (as pointed out in the related work), this has typically been focused on non-HRI scenarios and used to improve robot/environment dynamics models.
* The inclusion of the s2r-Oracle model in evaluation is very insightful. The effort required to perform multiple real-world policy rollouts with human partners as in i-S2R may be overly prohibitive, but if this process only needs to be performed once to learn a sufficient human model which can then be used to train new policies from scratch, this would go a long way to making such a methodology more broadly applicable. However, this does beg the question: how does this compare to collecting human data up front (with another human partner) and learning the human behavior model with imitation learning? It is not entirely clear to me that the robot is required to collect human-data in this table tennis scenario. However, in cases where the robot’s presence is required for human data collection (e.g., there would be significant distribution shift when a robot is present vs a human), such an approach seems worthwhile.
* The results are quite impressive! The included video showing the robot executing its table tennis policy with a human opponent for up to 150 hits in a single rally leaves an impression.

Weaknesses:
* The small sample size in human participants somewhat limits the conclusions that can be drawn from the experimental results. While this is clearly acknowledged in the paper and is somewhat inherent to human-robot interaction, the outlier that is the advanced player introduces many questions about under what conditions i-S2R is preferable over standard S2R.
* No conjecture seems to be made about why i-S2R performs worse for the advanced player. Is there possibly some aspect of their play that may have changed? Are there any simulation reward curves that might indicate a failure during the training process during the latter iterations? Did the fine tuning process go awry?
* The fine-tuning in the real-world process doesn’t necessarily make sense to me. Presumably, one needs to fine-tune the policy in the real-world because there is some slight distribution shift between simulation and real? But then why take the fine-tuned policy back into simulation? It seems like this would constantly be introducing distribution shift into the training process and make things more difficult. Prior works like [2] attempt to adjust the simulation models to make it more realistic, thus reducing the distribution shift on every iteration. But in this case, I’m not sure what the fine-tuning is accomplishing or why one would want to bring it back to simulation.
* It would have been interesting to train a single s2r-oracle policy with the aggregated data from all players. Would it generalize better than the individual policies given that the behavior model represents a wider distribution of player data? It is not entirely clear to me why individual policies were trained for each player rather than mixing and matching data.
* Minor: The right side of Fig. 2 is a bit difficult to understand. After reading the textual description of the different variants (Sec. 6) and spending a significant amount of time trying to understand the figure, I think I understand it, but it is not intuitive at first glance.


**Summary Of Recommendation:**

This paper addresses an important research problem, proposes an interesting method, is well-written, and invites interesting discussion about the results. While aspects of the method are not entirely novel and the small sample size in human participants yields a lot of variance in the data, I think this paper is of interest to the broader robotics -- and in particular HRI -- community given the challenge in leveraging reinforcement learning when humans are involved. The trained table tennis policies are also quite impressive and demonstrate some significant steps forward, particularly with successful play against novel opponents in the real-world.

---

> ### Author Response · Authors · 2022-08-24
> **Authors response to Reviewer WKtY (1/2)**
>
> **Comment:**
>
> Thank you for your helpful and detailed feedback. We appreciate the time and effort you took to review our work. We are also glad that you find i-S2R interesting and appreciated our inclusion of S2R-Oracle and real world results.
>
> **On method naming**
>
> Please note in response to reviewer LWg8’s feedback we re-named our baselines as follows
> * S2R-out-of-sim → S2R
> * S2R → S2R+FT
> * S2R-Oracle-out-of-sim → S2R-Oracle
> * S2R-Oracle → S2R-Oracle+FT
>
> **On the human-subject experiment size**
>
> While our main method only requires 6 hours of in-person fine-tuning, our evaluation involves 3 different fine-tuning runs (our method and two baselines), and we evaluate 10 different checkpoints for each person. In aggregate, adding each person requires an additional 20 hours of human and robot time. This is before we take into account rest periods for humans and any robot downtime that we might encounter. Our experiments already exceeded over 100 hours of human and robot time in aggregate, and it is difficult and expensive to increase this number in a significant way. Further, since we have only a single robot setup, we have to run our experiments one person at a time, and we are only able to obtain results for a maximum of two people per week (assuming no downtime at all). We do agree that a larger sample size would have been preferable, however given the above constraints we did our best to compensate for the relatively small sample size by making our evaluations as comprehensive and unbiased as possible.
>
> **On the advanced player’s data**
>
> We looked into simulated training curves and did not observe any systematic differences that might explain the surprising performance of i-S2R for the advanced player.
>
> One possible explanation is that the i-S2R policy for the advanced player played fast. During evaluations, we observe the initial robot return is fast with top spin, likely due to a combination of changes in the behavior model from iteration 1 to 2 and 3 and inherent randomness in the training process. In response, the advanced player returns the ball even faster, also with top spin. This appears challenging for the robot to return. During evaluation most of the errors are made by the robot, where the rally ends with the ball going over the human player’s end of the table. This suggests that fine-tuning was not able to adjust in time to the top spin and fast speed of play, causing the robot to hit over the table. One way to mitigate this would be to model spin in simulation, so the policy could learn to respond to spin throughout training, not just during fine-tuning.
>
> We have updated Section 6 (1) and the limitations section of the paper to reflect this discussion. Please see the revised version attached with changes in blue.
>
> **Why is the fine-tuned model from the real world brought back to simulation?**
>
> There are two reasons for bringing the fine-tuned model from the real world back to simulation. First, the policy has been trained on the most up to date human behavior. As a result it is likely better adapted to play with the updated human behavior model than the last set of policy weights from simulated training which were trained on the prior human behavior model.
>
> Second, there are aspects of our real world system which we have not been able to model accurately in simulation on top of the challenges in modeling human behavior: Our vision system does not detect spin, there are calibration defects, variability in estimated delays, conditions of the surface materials, wear and tear (of table tennis balls), physical robot properties mismatched with the simulated robot. Therefore, our policies are subject to a sim2real gap and real world fine tuning adapts the policy to real world conditions. When we transfer the fine tuned policy weights back to simulation we observe that some adaptation to real world conditions persists from iteration to iteration, reducing the adaptation time in subsequent fine tuning iterations.
>
> However it is possible that this approach makes training in simulation more difficult. It would be interesting to compare our approach with a variant in which the policy weights are not transferred back to simulation from the real world. Instead training in simulation would continue using the latest policy weights from the previous iteration but using the latest human model after real fine-tuning. We leave this to future work.
>
> **Zip File:**
>
> /attachment/c45c7b83ae86257b107288e2310e412c1e10645f.zip

---

> > ### Author Response · Authors · 2022-08-24
> > **Authors response to Reviewer WKtY (2/2)**
> >
> > **Comment:**
> >
> > **On the differences between S2R-Oracle and S2R-Oracle-out-of-sim**
> >
> > We hypothesize that the difference between S2R-Oracle+FT (previously S2R-Oracle) and S2R-Oracle (previously S2R-Oracle-out-of-sim) is mainly attributable to a persistent sim-to-real gap which is overcome through fine-tuning. In the presence of the sim-to-real gap, fine-tuning makes a significant difference to performance. Were the sim-to-real gap to be smaller, we hypothesize the impact of fine-tuning would be less.
> >
> > However, we do think that the human behavior model makes an important difference. S2R-Oracle+FT achieves comparable performance to i-S2R on average, yet only needs 35% of the real world fine-tuning time. We think this is due to the human behavior model. S2R-Oracle+FT has access to the final human behavior model that it took i-S2R three iterations to generate. Further, the comparable performance of S2R-Oracle+FT and i-S2R validates our hypothesis that the difficulty of defining a good human behavior model a priori for human-robot cooperative rallies was limiting performance. Finally, the fact that S2R+FT (previously S2R) does not perform as well as i-S2R or S2R-Oracle+FT also indicates the human behavior model is important.
> >
> > **Are there statistical significance results for the comparisons between the different S2R variants?**
> >
> > The un-normalized mean rally length for i-S2R, S2R+FT and S2R-Oracle+FT is not statistically significantly different, since the 95% confidence intervals overlap (see Appendix J, Figure 8 c). However, the histogram of rally lengths for i-S2R and S2R+FT shows that a large fraction of the rallies for S2R+FT are shorter (i.e. less than 5), while i-S2R achieves longer rallies more frequently. This suggests i-S2R yields policies that are more fun to play with on average.
> >
> > When rally length is normalized to account for differences in skill level between players (see Appendix J, Figure (a)), the mean rally length for S2R-Oracle+FT is statistically significantly higher than S2R+FT, although the difference is small.
> >
> > Finally, when the advanced player is excluded, the mean rally length (normalized and unnormalized) for i-S2R is statistically significantly higher than S2R+FT and the difference is large. The mean rally length for i-S2R and S2R-Oracle+FT are not statistically significantly different (see Appendix J, Figure 9 for un-normalized data).
> >
> > **What distinguishes a beginner, intermediate, and advanced player?**
> >
> > We group players according to empirical skill (i.e. how they actually played) as opposed to using self-reported skill because non-professional players' perception of their skill level may not be well calibrated across players (see Appendix H).
> >
> > The metrics we used to measure empirical skill were rally length statistics (mean, 75th percentile, and max) averaged over all models and evaluations (see Appendix I, Table 8).
> >
> > Beginners have a mean rally length ~7 - 10.5 hits, 75th percentile from 9-13 hits, and maximum rally lengths of <90 hits. Intermediate players have a mean rally length of ~14 - 17 hits, 75th percentile from 15-18 hits, and maximum rally lengths from 100 - 200 hits. The advanced player had a mean rally length of 19.4, 75th percentile of 22 hits and a maximum rally length of 345 hits.
> >
> > **Training a single oracle policy with aggregated data from all players**
> >
> > In this work we focused on the task of playing cooperatively with a single human opponent. We repeated the training process independently with multiple humans to test the replicability of our method. However this would be very interesting to try in future work, thank you for the suggestion.
> >
> >
> > **Zip File:**
> >
> > /attachment/4886541c15ae3f035b1623d4e6eb26ce17b35e57.zip

---

> > > ### Comment · Reviewer_WKtY · 2022-08-25
> > > **Response to Author**
> > >
> > > Thank you for the detailed response.
> > >
> > > The additional information in the paper regarding the advance player’s results are useful for the reader. I would also suggest adding something related to the explanation given here for bringing the fine-tuned real-world model back to simulation on each iteration. It is easy for one to imagine scenarios in which the fine-tuning captures dynamics that are not reproduced in simulation, thus causing a constant distribution shift and disrupting training efficiency. So any explanation regarding this would be beneficial to readers.
> > >
> > > Regarding statistical significance, I would recommend indicating some of these significance results in the paper. The term “significantly” is used frequently in the paper but it is not clear whether this is referring to a colloquial significance or statistical significance. This should be clarified in the manuscript.

---

> > > > ### Author Response · Authors · 2022-08-26
> > > > **Thank you for your additional comments.**
> > > >
> > > > Thank you for your additional comments. These are great suggestions.
> > > >
> > > > We will add the explanation about bringing the fine-tuned real-world model back to simulation to the final version of this work.
> > > >
> > > > We note that almost all of the significance results are already included in the paper although they are in the appendix due to space constraints. Additionally, we will add the discussion here on statistical significance and a chart on all players excluding the advanced player to the final version of this work. Finally, we will clarify our wording throughout the paper to make it clear whether we are using "significant" statistically or colloquially.

---

### Comment · Area_Chair_Xy9z · 2022-08-25
**Authors have responded in detail;  further thoughts & discussion?**

Dear reviewers -- the authors have provided detailed responses to the reviews.  Please give these responses a careful read, and provide further feedback & engagement with the authors now, as the author/reviewer discussion/rebuttal phase ends Aug 27 at 11:59 PM Pacific.
Your participation greatly contributes to the overall quality and value of the review process, and so it is very much appreciated! -- your Area Chair

---

### Meta-Review · Area_Chair_Xy9z · 2022-08-14

**Recommendation:** Accept (Oral)
**Confidence:** 5

**Metareview:**

The paper proposes iterative sim-to-real (i-S2R) to train robotic policies that are proficient at
interacting with humans upon deployment.  Collecting data in simulation requires a human model,
while collecting interaction data in the real world requires a policy for the human to interact
with.  This chicken-and-egg problem is resolved via an iterative approach, bootstrapPING from a simple
model of human behavior and alternatING between training in simulation and deploying in the real
world.

The paper has recommendations of 2x strong accept, 2x accept.
The reviewers (and the area chair!) believe this paper to be quite interesting.
The paper could be considered for an Oral presentation and/or best paper, given the fundamental and important nature of the problem being tackled.

Strengths:
- well motivated problem
- well written
- impressive results:  table tennis on 8-DOF robot
- very relevant to human-robot interaction
- fairly novel & interesting; prev work on iterative sim/world data collection is usually non-HRI
- good use of s2r-Oracle model in evaluation

Weaknesses:
- may not transfer to other tasks, which have different attributes (true, but the pipeline itself is pretty general)
- possibly strong assumption for the strong human motion prior; humans also change over time (counterargument: already works remarkably well given how simple the model actually is)
- further discussion on the advanced player’s surprising data would improve the paper (now addressed)
- small sample size (5) for human participants (now an acknowledge limitation)
- metric for evaluation should be described early on (now addressed)
- validity of baseline S2R method is not clear (addressed in rebuttal)

---

> ### Author Response · Authors · 2022-08-24
> **Authors response to Area Chair (1/2)**
>
> **Comment:**
>
> Thank you for your helpful comments and synthesis of the reviews. Please see our responses below to the weaknesses you highlighted. We also attach a revised version of our paper (changes highlighted in blue) which reflects our comments here and in our individual responses to reviewers.
>
> **On method naming**
>
> Please note in response to reviewer LWg8’s feedback we re-named our baselines as follows
> * S2R-out-of-sim → S2R
> * S2R → S2R+FT
> * S2R-Oracle-out-of-sim → S2R-Oracle
> * S2R-Oracle → S2R-Oracle+FT
>
> **Method may not transfer to other tasks.**
>
> We would like to point out that our primary contribution in the paper is the i-S2R algorithm: an online, iterative approach that alternates between simulated training, real world training and data-collection, and behavior modeling. We chose a specific approach to model human behaviors for our task, and we do not claim that this particular approach of human behavior modeling will work for any arbitrary task. However, we believe that our overall pipeline of bootstrapping from a small amount of human data, and then iteratively refining both our human behavior models and learned robot policies in an online fashion can be applied to a broad range of tasks, and take into account the various nuances of those tasks. The two characteristics a human behavior model needs to work with i-S2R are (a) it can be updated using human data that is gathered whilst a human or humans are interacting with a robot, and (b) the model can be used to sample human behavior in simulation. We therefore think a large class of human behavior models are compatible with the i-S2R algorithm. Perhaps the most general way to build human models would be to use a powerful generative model (such as an autoregressive, decoder-only transformer), and this direction would be interesting to explore in future work. We have updated the introduction to reflect this discussion.
>
> **Assumption for human motion prior**
>
> We endeavored to accurately model the ball in simulation, for example we have measured the physical parameters of the ball, paddle, and table using standard methods, and the simulator is fixed to those values. However, we know that our model is not perfect. For example, we have not modeled spin or physical wear and tear of the balls. These unknown parameters do introduce a sim-to-real gap and we observe a drop-off in performance between simulation and the real world (see Figure 5). Our hypothesis is that fine-tuning is able to adapt reasonably well to these inaccuracies, although we think that were the sim-to-real gap to be reduced, performance would improve.
>
> While it is true that i-S2R assumes that the human behavior model can be represented and sampled from in simulation, we note that our model is quite simple. We reduce human ball trajectories into a set of position and velocity bounds which we sample from uniformly. Further, the distribution does not need to be expressed analytically, all that is needed is the ability to sample from it in simulation. A generative model (such as an autoregressive, decoder-only transformer) is also compatible with i-S2R, and this direction would be interesting to explore in future work.
>
> For these reasons we believe that the human behavior model requirement is not especially strict and that i-S2R is robust to some inaccuracies in the behavior model. Assessing the sensitivity of the method to different human behavior models of varying degrees of fidelity is an interesting direction for future work.
>
> **Humans also change over time**
>
> We completely agree that humans change over time and that this is important in collaborative robotics. We believe that i-S2R is able to adapt to a changing human — this is precisely what the multiple rounds of real world fine-tuning in i-S2R facilitate. Our focus in this paper was on how to address the fact that a human’s policy changes as the robot gets better. However we think i-S2R could equally be applied in settings in which a human’s policy changes due to other factors (such as learning or forgetting). Each time the human policy changes an additional iteration could be performed.
>
> We do note the possibility of humans changing during training, specifically of them getting better, in our limitations section (lines 321 - 326 in the original pdf). Since our focus was on evaluating the robots we made efforts to exclude the possibility of humans changing from our evaluations.
>
> Finally, we think that the human and the robot do find a good joint policy. Evidence for this is the cross evaluations (Figure 7). Here we see that when a policy is evaluated by a player that wasn’t involved in the training of the policy performance drops, suggesting some amount of co-adaptation between the robot and player. Qualitatively this was also the experience of our test subjects. They describe learning to adapt to the policy’s style during fine-tuning.
>
>
>
> **Zip File:**
>
> /attachment/5817446f05e27efb07b4bd4b93419149366429d5.zip

---

> > ### Author Response · Authors · 2022-08-24
> > **Authors response to Area Chair (2/2)**
> >
> > **Comment:**
> >
> > **Further discussion on the advanced player’s surprising data**
> >
> > We looked into simulated training curves and did not observe any systematic differences that might explain the surprising performance of i-S2R for the advanced player.
> >
> > One possible explanation is that the i-S2R policy for the advanced player played fast. During evaluations, we observe the initial robot return is fast with top spin, likely due to a combination of changes in the behavior model from iteration 1 to 2 and 3 and inherent randomness in the training process. In response, the advanced player returns the ball even faster, also with top spin. This appears challenging for the robot to return. During evaluation most of the errors are made by the robot, where the rally ends with the ball going over the human player’s end of the table. This suggests that fine-tuning was not able to adjust in time to the top spin and fast speed of play, causing the robot to hit over the table. One way to mitigate this would be to model spin in simulation, so the policy could learn to respond to spin throughout training, not just during fine-tuning.
> >
> > We have updated Section 6 (1) and the limitations section of the paper to reflect this discussion.
> >
> > **Small sample size (5) for human participants.**
> >
> > While our main method only requires 6 hours of in-person fine-tuning, our evaluation involves 3 different fine-tuning runs (our method and two baselines), and we evaluate 10 different checkpoints for each person. In aggregate, adding each person requires an additional 20 hours of human and robot time. This is before we take into account rest periods for humans and any robot downtime that we might encounter. Our experiments already exceeded over 100 hours of human and robot time, and it is difficult and expensive to increase this number in a significant way. Further, since we have only a single robot setup, we have to run our experiments one person at a time, and we are only able to obtain results for a maximum of two people per week (assuming no downtime at all). We do agree that a larger sample size would have been preferable, however given the above constraints we did our best to compensate for the relatively small sample size by making our evaluations as comprehensive and unbiased as possible.
> >
> > **Metric for evaluation should be described early on.**
> >
> > We have clarified this in the introduction. Please see the revised version of our submission attached with this comment.
> >
> > **Validity of baseline S2R method is not clear**
> >
> > As we describe in the related work section, the most common approach to sim-to-real in robotics is zero-shot transfer from simulation [1, 3, 5, 6, 7, 8] and we think this is the standard method. This baseline is included in our results as “i-S2R-out-of-sim” and we acknowledge that this labeling is perhaps confusing. To address this, we have renamed “i-S2R-out-of-sim” to “S2R”.
> >
> > However, we wished to compare i-S2R with a stronger baseline than the standard method, hence why we mainly compare with sim-to-real plus fine-tuning. This was labeled as “S2R” in our results and we have re-named it to “S2R+FT” to emphasize that fine-tuning is involved.Our implementation of S2R+FT adopts the approach taken in [24].
> >
> > Comparing i-S2R and S2R+FT is a more like-for-like comparison than with S2R because both i-S2R and S2R+FT have the same real world fine-tuning budget. In comparison, S2R does not have any real world training.
> >
> > We have updated all the charts and text in the revised version of our submission to reflect these changes.
> >
> > [1] X. B. Peng, M. Andrychowicz, W. Zaremba, and P. Abbeel.  Sim-to-real transfer of robotic control with dynamics randomization, 2018
> >
> > [3] M. Andrychowicz, B. Baker, M. Chociej, R. Jozefowicz, B. McGrew, J. Pachocki, A. Petron, M. Plappert, G. Powell, A. Ray, J. Schneider, S. Sidor, J. Tobin, P. Welinder, L. Weng, and W. Zaremba. Learning dexterous in-hand manipulation, 2020
> >
> > [5] J. Lee, A. Dosovitskiy, D. Bellicoso, V. Tsounis, V. Koltun, and M. Hutter. Learning agile and dynamic motor skills for legged robots, 2019
> >
> > [6] X. B. Peng, E. Coumans, T. Zhang, T. E. Lee, J. Tan, and S. Levine.  Learning agile robotic locomotion skills by imitating animals, 2020
> >
> > [7] F. Sadeghi and S. Levine.  CAD2RL: real single-image flight without a single real image, 2017
> >
> > [8] A. Loquercio, E. Kaufmann, R. Ranftl, M. Muller, V. Koltun, and D. Scaramuzza.  Learning high-speed flight in the wild, 2021.
> >
> > [24] L. M. Smith, J. C. Kew, X. B. Peng, S. Ha, J. Tan, and S. Levine. Legged robots that keep on learning, 2021
> >
> >
> > **Zip File:**
> >
> > /attachment/dcb027c57e1f58d9d25beaec74be402e736dd921.zip